

# Association between non-alcoholic fatty liver disease and arterial stiffness measured by brachial-ankle pulse wave velocity: a cross-sectional population study

Yujie Wang[1,*], Zhicheng Fang[2,*], Qiuyue Fu[1], Dongai Yao[3] and Xiaoqing Jin[1,3]

[1] The Emergency Center, Zhongnan Hospital of Wuhan University, Wuhan, Hubei, China
[2] Emergency Department, Taihe Hospital of Hubei University of Medicine, Shiyan, Hubei, China
[3] Physical Examination Center, Zhongnan Hospital of Wuhan University, Wuhan, Hubei, China
[*] These authors contributed equally to this work.

Corresponding authors
Dongai Yao, yellowcattjzx@sina.com
Xiaoqing Jin, redjin@whu.edu.cn

## ABSTRACT

**Background.** Non-alcoholic fatty liver disease (NAFLD) is strongly linked with metabolic syndrome and atherosclerotic cardiovascular diseases (ASCVDs). This study aimed to assess the feasibility of using brachial-ankle pulse wave velocity (baPWV), a non-invasive technique, to monitor atherosclerosis (AS) in NAFLD patients and to evaluate the AS risk in various sub-populations of NAFLD patients.

**Materials and methods.** A cross-sectional study was conducted with 4,844 participants, enrolled from January 1, 2019, to December 31, 2021, at the Physical Examination Center of Zhongnan Hospital of Wuhan University. Participants were aged 18 to 88 years. According to the main points of the ultrasonic diagnosis of NAFLD, the ultrasonic image report was made for the subjects. AS is defined as baPWV $\geq$ 1,400 cm/s. We used multiple logistic regression analysis to explore the relationship between NAFLD and AS, and multiple linear regression analysis to explore the correlation between NAFLD and baPWV by modeling. Subgroup analysis was performed based on age and gender to adjust for confounding bias and complete sensitivity analysis.

**Results.** The prevalence of NAFLD was 38.3% in all participants, with 45.4% in men and 25.1% in women. Among the overall NAFLD population and male NAFLD patients, baPWV exceeded the diagnostic threshold for AS (1,419.70 $\pm$ 205.51, 1,429.71 $\pm$ 196.13) starting from the 45–55 age group. Through the analysis of the age-baPWV scatter plots and fitted lines, along with sensitivity analysis, it is recommended that male patients should start monitoring at 46 years old for AS using baPWV, while female patients should begin at 51 years old. NAFLD was associated with increased odds of AS (OR: 1.206, 95% CI [1.021–1.423], $P = 0.027$) after adjusting for confounders. NAFLD was independently positively correlated with baPWV (Model 2: $\beta = 0.086$, $\Delta R^2 = 0.006$, $P < 0.001$; Model 3: $\beta = 0.05$, $P < 0.001$). This positive correlation was also observed in both males and females (male: Model 2: $\beta = 0.081$, $\Delta R^2 = 0.005$, $P < 0.001$; Model 3: $\beta = 0.052$, $P = 0.001$; female: Model 2: $\beta = 0.088$, $\Delta R^2 = 0.006$, $P < 0.001$; Model 3: $\beta = 0.042$, $P = 0.02$).

**Conclusion.** NAFLD demonstrated an independent association with AS assessed via baPWV, an accessible non-invasive tool for early AS evaluation. Regular baPWV

monitoring is recommended for NAFLD patients > 45 years, with males and females initiating surveillance at 46 and 51 years, respectively. Study limitations, including potential biases in NAFLD diagnosis, gender distribution imbalances, and confounding variable interdependencies, necessitate further stratified population analyses.

# INTRODUCTION

Non-alcoholic fatty liver disease (NAFLD) is a common chronic liver condition associated with metabolic syndrome, affecting approximately 25% of the global population and steadily increasing (*Powell, Wong & Rinella, 2021*). NAFLD includes a range of disorders, from non-alcoholic fatty liver (NAFL) to non-alcoholic steatohepatitis (NASH), potentially progressing to fibrosis and cirrhosis (*Engin, 2017*). Evidence suggests that NAFLD is not only a liver-specific condition but also a significant predictor of atherosclerotic cardiovascular diseases (ASCVDs) (*Adams et al., 2017*; *Lebovics & Rubin, 2011*; *Santos et al., 2022*; *Targher, Byrne & Tilg, 2020*). ASCVDs are the leading cause of death among NAFLD patients, especially those with multiple metabolic disorders (*Ismaiel & Dumitraşcu, 2019*). Compared to individuals without NAFLD, those with NAFLD exhibit reduced vasodilation capacity, increased intima-media thickness (IMT), and a higher presence of vulnerable atherosclerotic plaques (*Taharboucht et al., 2021*), all closely associated with ASCVD (*Tang et al., 2015*). Therefore, it is essential to explore the relationship between NAFLD and ASCVDs and to monitor ASCVDs occurrence and severity in NAFLD patients at an early stage.

Atherosclerosis (AS), characterized by impaired arterial elasticity, is an early manifestation of ASCVDs. Timely detection of decreased arterial elasticity in NAFLD patients is critical for monitoring and preventing AS progression. Previous studies have primarily used vascular ultrasound, coronary computed tomography angiography (CCTA), and coronary angiography (CAG) to detect IMT and atherosclerotic plaque. These methods are invasive, require high-quality equipment, and need specialized operators, complicating AS diagnosis (*Cho et al., 2018*; *Koulaouzidis et al., 2021*; *Lee et al., 2018*; *Wong et al., 2021*).

Brachial-ankle pulse wave velocity (baPWV) is widely recognized as a valuable non-invasive tool for assessing arterial health and providing early indications of atherosclerotic changes. In comparison, baPWV provides a clear advantage for early detection. BaPWV can identify hemodynamic alterations linked to endothelial dysfunction and vascular remodeling (*Munakata, 2016*). Due to its ease of use and reliability, baPWV is expected to be adopted widely in community hospitals and clinics (*Zhu et al., 2015*). A meta-analysis indicated that China and other Asian countries are characterized by high pulse wave velocity, and the differences in baPWV are not significant, regardless of the measurement method, device manufacturer, or optical path measurement used (*Lu et al., 2023*). A meta-analysis indicated that an increase of 1 m/s in baPWV was associated with 12%,

13%, and 6% increases in total cardiovascular (CV) events, CV mortality, and all-cause mortality (*Vlachopoulos et al., 2012*). However, due to variations in research design, study populations, and detection methods, there are doubts about whether baPWV is suitable for the detection of arterial elasticity in patients with NAFLD. A study of the association between nonalcoholic fatty liver disease and subclinical atherosclerosis in obese Chinese adults indicated that NAFLD was not associated with baPWV (*Liu et al., 2017*). In another prospective study, both baPWV and CIMT elevation were independently associated with the development of NAFLD (*Xin et al., 2020*).

This study aims to evaluate the association between NAFLD and AS using baPWV. Utilizing a large sample size with extensive data, we seek to gain a comprehensive understanding of the systemic effects of NAFLD and to highlight the potential of baPWV as a non-invasive technique for early AS detection in diverse NAFLD populations.

## MATERIAL AND METHODS

### Subjects and study design

This cross-sectional study was conducted at Zhongnan Hospital of Wuhan University, enrolling 5,439 individuals who completed physical examinations at the Physical Examination Center from January 1, 2019, to December 31, 2021. The study was approved by the Ethics Committee of Zhongnan Hospital of Wuhan University (NO.2021114) and it was granted an exemption from informed consent. Ultimately, 4,844 individuals were included, comprising 3,156 males and 1,688 females, aged 18 to 88 years.

Exclusion criteria were: (1) Missing important basic information (age, gender, biochemical indicators, *etc.*); (2) Presence of other known causes of liver disease, including viral hepatitis, drug-induced hepatitis, and alcoholic hepatitis; (3) Excessive alcohol consumption (>140 g/week for men and >70 g/week for women); (4) Self-reported history of cardiovascular diseases; (5) Parenteral nutrition; (6) Intake of specific medications (amiodarone, methotrexate, tamoxifen, corticosteroids, valproate, and antiretroviral drugs); (7) Pregnancy; (8) Missing or unreasonable baPWV data (see Fig. 1).

### Sample size calculation

The sample size for this retrospective study was calculated based on the primary analysis using multivariate logistic regression to assess the association between NAFLD and AS, with an anticipated odds ratio (OR). The calculation considered 14 independent variables, including NAFLD, to ensure adequate power for detecting significant associations.

The formula for sample size estimation in logistic regression was applied:

$$n = \frac{(Z_{1-\alpha/2} + Z_{1-\beta})^2}{\ln(OR)^2 \cdot p \cdot (1-p) \cdot (1-R^2)}$$

where:

- $Z_{1-\alpha/2} = 1.96 \, (two-tailed \; \alpha = 0.05)$
- $Z_{1-\beta} = 0.84 \, (power = 80\%)$
- OR $= 1.5$ (clinically meaningful effect size based on prior studies),
- $p = 0.3$ (prevalence of NAFLD in the general population),

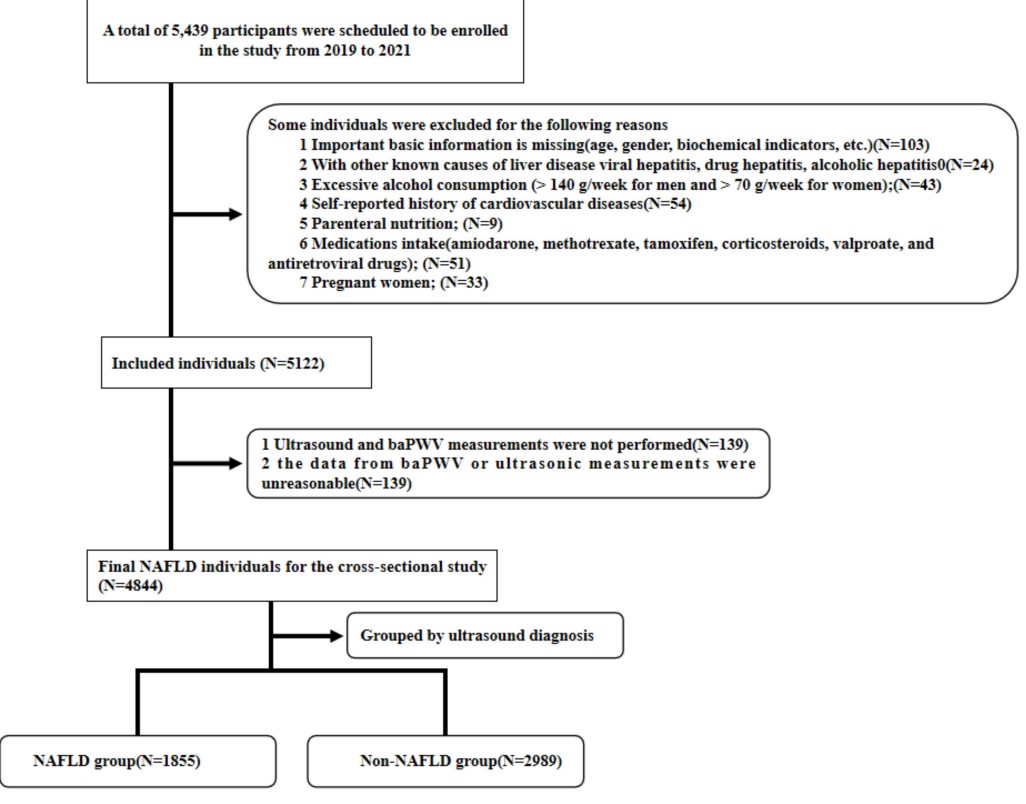

**Figure 1  Flow chart of the participant selection procedure.**

- $R^2 = 0.2$ (variance explained by covariates).

Substituting values:

$$n = \frac{(1.96 + 0.84)^2}{\ln(1.5)^2 \cdot 0.3 \cdot (0.7) \cdot (0.8)} \approx 680.$$

To account for potential confounding and ensure robustness for subgroup analyses (gender-specific strata and age-stratified ANOVA), the sample size was increased by 20%, resulting in a minimum required sample of 816 participants.

Validation for secondary analyses

1. Multivariate linear regression (12 variables): Adhering to the rule of $n \geq 50 + 8kn$, where $k = 12$, the minimum sample size was 146. Our calculated sample ($n = 816$) exceeded this threshold.

2. ANOVA for age-stratified baPWV: With six age groups and four comparison subgroups, a minimum of 30 participants per cell was ensured ($6 \times 4 \times 30 = 7,206 \times 4 \times 30 = 720$), further validating the adequacy of n=816\.

After adjusting for 10% potential data incompleteness, the target sample size was 900 participants.

A total of 4,844 participants were enrolled in our study, including 1,855 with NAFLD and 2,989 without NAFLD. It clearly meets the requirement of sample size.

## Quality control

Data collectors were trained on the electronic medical record system data collection process, using a standardized approach to minimize errors and biases. We selected participants from a continuous time period to avoid selection bias. Missing data were identified and excluded from the analysis. In both the logistic regression and multiple linear regression analyses, we included all confounders in the models and reported the adjusted results. Additionally, we conducted subgroup analyses for males and females, as well as sensitivity analyses for both the multiple linear regression and age-baPWV trend analyses. These steps were taken to minimize the impact of confounding factors on the study outcomes. Our research data is obtained according to a prescribed process. The data is clearly defined and can be collected and analyzed in other organizations as well. Therefore, this study is reproducible.

## Diagnosis of NAFLD

Liver biopsy remains the gold standard for NAFLD diagnosis. However, its clinical utility is limited by ethical concerns, high costs, invasiveness, sampling variability, and poor reproducibility (*Guan, Chen & Xu, 2022*). Additionally, the procedure carries notable risks, including a 0.2% mortality rate and an 84% incidence of pain, which contribute to low patient acceptance (*Khalifa & Rockey, 2020*; *Petzold, 2022*). Ultrasonography is the most widely used imaging technique for assessing hepatic steatosis and liver abnormalities due to its simplicity, cost-effectiveness, high reproducibility, and excellent patient tolerance. It has a sensitivity range of 53–76% and specificity of 76–93% for the detection of hepatic steatosis. When moderate-to-severe steatosis is suggested by sonographic signs (*e.g.*, reduced diaphragm visibility due to increased echogenicity), the diagnostic probability approaches 100% (*Ezenwuba & Hynes, 2024*; *Petzold, 2022*). NAFLD was diagnosed based on the Practice Guideline for the Diagnosis and Management of NAFLD by the American Association for the Study of Liver Diseases (*Chalasani et al., 2018*) and the guidelines recommended by the Asia-Pacific Working Party (*Farrell et al., 2007*). All individuals were categorized into either the NAFLD or non-NAFLD group using ultrasound, performed independently by two skilled ultrasound operators who were blind to the participants' clinical manifestations and biological test results to ensure an unbiased evaluation. If discrepancies arose between the evaluation results, a third professional operator would be consulted for a re-evaluation using the same criteria. The final imaging diagnosis was determined by considering the results of all three operators collectively. The imaging criteria for the diagnosis of NAFLD were met when the ultrasound imaging met the following characteristics: (a) The hepatic near-field echo (bright) showed diffuse enhancement compared to the kidney; (b) Intrahepatic ductal structures are unclear; (c) Attenuation of the liver's far-field echoes (*Takeuchi et al., 2012*). The clinician determines the diagnosis of NAFLD by combining inclusion and exclusion criteria with the patient's liver ultrasound results.

## Clinical characters and laboratory measurements

Clinical data of the participants were obtained through an electronic medical record review at the health examination center. The main non-imaging biomarker-based scoring
systems for liver fat composition currently include FibroTest (FT), ActiTest (AT), and SteatoTest (ST). FT is composed of serum α2-macroglobulin, apolipoprotein A1, haptoglobin, total bilirubin, and γ-glutamyl transferase. AT builds upon FT by adding alanine aminotransferase (ALT). ST contains the same six components as FT and AT, with the addition of body mass index (BMI), serum cholesterol, triglycerides (TG), and glucose, and further adjusts for age and sex (*Munteanu et al., 2016*). A review of diagnostic methods for NAFLD points out that elevated ALT has limited sensitivity and specificity for detecting NASH. And of these systems, only ST has demonstrated superior diagnostic accuracy when validated against liver biopsy (*Pirmoazen et al., 2020*). In a scientific statement, the American Heart Association notes that ASL and ALT levels may be normal in people with NAFLD (*Duell et al., 2022*). In the context of clinical practice in primary hospitals in China, the biomarkers used in FibroTest are less commonly employed, while the additional markers in SteatoTest are more widely incorporated into routine clinical tests. So the data we chose included age, sex, smoking status, alcohol consumption, physical activity, BMI, total cholesterol (TC), TG, high-density lipoprotein (HDL), low-density lipoprotein (LDL), uric acid (UA), fasting blood glucose (FBG), hypertension, history of medication use, and cardiovascular diseases. Definitions of each indicator are provided in the supplementary materials.

## Arterial stiffness measurement and AS severity

BaPWV measurements were conducted using the automated AS determinator VP-1000, manufactured by the Japan Colin Company, Tokyo, Japan. The medical personnel performing the baPWV assessments were blind to the participants' clinical presentations and biological test outcomes. One limb, either left or right, was chosen for measurement. The cuff was positioned on the upper arm and ankle of the chosen limb, enabling simultaneous measurement of the pulse waves. The disparity in pulse wave initiation times was adjusted for the distance factor. The higher value obtained from the baPWV measurements was used as the definitive result. According to the Chinese expert consensus draft on early vascular disease detection, AS is defined as baPWV $\geq$ 1,400 cm/s, while baPWV <1,400 cm/s indicates normal arterial elasticity. Specifically, baPWV values between 1,400 cm/s and 1,600 cm/s, 1,600 cm/s and 1,800 cm/s, and exceeding 1,800 cm/s indicate mild, moderate, and severe AS, respectively.

## Grouping method

In the age models, individuals were stratified into five categories. BaPWV measurements were conducted across different age groups in both healthy individuals and NAFLD patients to assess the significance of baPWV in monitoring AS among NAFLD populations of varying ages. Gender subgroup analysis was also performed to reduce bias due to the unbalanced ratio of male to female participants. In the model of aggregate confounding factors, we included metabolic risk factors associated with ASCVDs and NAFLD, including hypertension, abnormal FBG, HDL, LDL, TC, TG, and UA by referring to modified National Cholesterol Education Program Adult Treatment Panel III criteria (*Fan et al., 2007*) with modifications recommended in the latest American Heart Association/National Heart, Lung, and Blood Institute Scientific Statement (*Grundy et al., 2005*).

We established three multiple linear regression models to investigate the relationships between confounders and baPWV. In Model 1, age, sex, smoking, alcohol consumption, physical activity, and BMI were included. Model 2 added NAFLD to assess its association with baPWV after adjusting for the variables in Model 1. In Model 3, metabolic and cardiovascular risk factors related to the occurrence of NAFLD and AS, including high TC, high TG, high UA, high FBG, hypertension, and low HDL, were further included as covariables. We also performed a gender subgroup analysis and sensitivity analysis.

## Sensitivity analysis

We replaced missing continuous variables with the mean values and missing categorical variables with the mode, resulting in a complete dataset. The imputed dataset was then subjected to sensitivity analysis, and the multiple linear regression model was reapplied to verify the relationship between NAFLD and baPWV across different populations. Additionally, age-baPWV scatter plots and fitted curves were generated for the entire NAFLD population, the non-NAFLD group, and males and females within the NAFLD group. This approach enabled us to investigate the sensitive age cut-off points for using baPWV to monitor the occurrence of AS in different populations.

## Propensity score matching and gender balance analysis

In order to balance confounding factors such as large gender ratio gap, we conducted propensity score matching (PSM) and a subsample analysis of gender ratio balance. The propensity score was estimated *via* logistic regression, with NAFLD status as the dependent variable and the following covariates included: sex, age, BMI, smoking, drinking, exercise, hypertension, FBG, HDL-C, TG, LDL-C, UA and TC. A caliper width of 0.2 standard deviations of the propensity score logit was applied to restrict matches to participants with similar baseline characteristics. A gender-balanced subsample was constructed by randomly selecting 423 males (matching the number of females in the NAFLD group) through stratified sampling with a fixed random seed (seed = 123). The final subsample comprised 846 NAFLD patients (50% male) and 2,989 non-NAFLD controls.

## Statistical analysis

To ensure that the data met the assumption of normality and to select the appropriate statistical methods, we conducted normality tests on all continuous variables using the Shapiro–Wilk test. For each variable, the *p*-value from the Shapiro–Wilk test was calculated. Based on the results of the normality test, we chose parametric statistical methods suitable for normally distributed data for subsequent analysis. We used box plots for outlier detection, a visualization method where outliers typically lie outside the "whiskers" of the box plot, indicating data points beyond 1.5 times the interquartile range (IQR). Participants with identified outliers were excluded from the analysis. Continuous variables were expressed as the mean and standard deviation (±SD), and the *t*-test was used for two-group comparisons. Categorical variables were reported as frequencies and percentages, and the chi-square test was used for ratio comparisons. ANOVA was used to compare baPWV across different age groups. *Post-hoc* multiple comparisons and trend test were included for ANOVA. Multivariate logistic regression analysis was used to assess the association between

NAFLD and AS. Multiple linear regression analysis was used to evaluate the relationship between different risk factors and baPWV in the three models mentioned above. Model assumptions were rigorously evaluated. Variance inflation factors (VIF) confirmed the absence of multicollinearity (all VIF < 3). Residual plots and Shapiro–Wilk tests validated normality and homoscedasticity in linear regression. Performed 1:1 nearest neighbor propensity score matching (PSM) using the MatchIt package in R for PSM. Covariate balance before and after matching was assessed using standardized mean differences (SMD) and variance ratios. An SMD threshold of < 0.1 was considered indicative of adequate balance. A $P$-value < 0.05 in a two-tailed test indicated statistical significance.

## RESULT

### Baseline characteristics of NAFLD and non-NAFLD populations

Of the 4,844 participants, 1,855 (38.3%) were diagnosed with NAFLD, including 1,432 (77.2%) men and 423 (22.8%) women. Compared to the non-NAFLD group, the NAFLD group had significantly higher average age, BMI, and proportions of males, smokers, drinkers, and individuals with high TC, high TG, high UA, abnormal FBG, hypertension and low HDL (all $P < 0.001$). The proportion of individuals engaging in physical exercise was significantly lower in the NAFLD group ($P < 0.001$) (Table 1). The average baPWV was also significantly higher in the NAFLD group, surpassing the threshold for AS (1,471.89 ± 294.90 *vs.* 1,371.55 ± 282.25, $P < 0.001$) (Table 1).

### NAFLD was independently associated with AS

The related risk factors for AS were shown in Table 2. The proportions of individuals aged ≥45 years, males, smokers, drinkers, and those with a lack of physical exercise, hypertension, diabetes, high FBG, high TC, high TG, high UA, and NAFLD were all significantly higher in the AS group compared to the non-AS group (all $P < 0.05$). Notably, these parameters in AS individuals align with the features observed in NAFLD patients. As the severity of arterial stiffness increased, the proportions of individuals with NAFLD increased. We further found that NAFLD was associated with increased odds of AS (OR: 1.206, 95% CI [1.021–1.423], $P = 0.027$) after adjusting for confounders (Fig. 2, Table S1). However, BMI did not show a significant association with the occurrence of AS ($P = 0.724$).

### Association between NAFLD and baPWV

Using multiple linear regression analysis, after adjusting for basic characteristics, NAFLD was positively and independently correlated with baPWV in Model 2 ($\beta = 0.086$, $P < 0.001$) (Fig. 3, Table S2). Furthermore, the inclusion of NAFLD significantly enhanced the predictive capability of baPWV ($\Delta R^2 = 0.006$, from Model 1 to Model 2). After incorporating metabolic indicators in Model 3, NAFLD still demonstrated an independent positive correlation with baPWV ($\beta = 0.050$, $P < 0.001$, $\Delta R^2 = 0.058$). Moreover, the magnitude of this positive correlation significantly exceeded the correlations between abnormal lipid metabolism indexes such as high TC and high TG with baPWV ($\beta = 0.034$, 0.041, all $P < 0.05$). In addition, the positive correlation also held in the gender subgroup. In Model 2 and Model 3 of the men, NAFLD was positively and independently correlated

**Table 1 Demographic and clinical characteristics of NAFLD group and non-NAFLD group.**

| Characteristics | Total (n = 4,844) | NAFLD (n = 1,855) | Non-NAFLD (n = 2,989) |
|---|---|---|---|
| Age (years)[*] | 49.39 ± 11.678 | 50.13 ± 11.07 | 48.93 ± 12.02 |
| BMI (kg/m$^2$ )[*] | 24.13 ± 3.09 | 25.91 ± 2.72 | 23.04 ± 2.78 |
| baPWV (cm/s)[*] | 1,409.97 ± 291.24 | 1,471.89 ± 294.90 | 1,371.55 ± 282.25 |
| Male n (%)[*] | 3,156 (65.2) | 1,432 (77.2) | 1,724 (57.7) |
| High TC n (%)[*] | 1,606 (33.2) | 738 (39.8) | 868 (29.0) |
| High TG n (%)[*] | 1,659 (34.2) | 1,030 (55.5) | 629 (21.0) |
| Low HDL n (%)[*] | 4,659 (96.2) | 1,748 (94.2) | 2,911 (97.4) |
| High UA n (%)[*] | 388 (8.0) | 255 (13.7) | 133 (4.4) |
| High FBG n (%)[*] | 316 (6.5) | 214 (11.5) | 102 (3.4) |
| Diabetes n (%)[*] | 162 (3.3) | 97 (5.2) | 65 (2.2) |
| Hypertension n (%)[*] | 825 (17.0) | 444 (23.9) | 381 (12.7) |
| Current Smoking n (%)[*] | 565 (11.7) | 282 (15.2) | 283 (9.5) |
| Drinking n (%)[*] | 326 (6.7) | 180 (9.7) | 146 (4.9) |
| Exercise n (%)[*] | 540 (11.1) | 92 (5.0) | 448 (15.0) |

Notes.

Abbreviation: BMI, Body mass index; baPWV, Brachial-ankle pulse wave velocity; TC, Total cholesterol; TG, Triglycerides; HDL, High-density lipoprotein; UA, Uric acid; FBG, Fasting blood glucose.

[*]$P < 0.001$.

Continuous variables are expressed as mean ± standard deviation, and categorical variables are expressed as counts and percentages. The inter-group comparison is based on ANOVA or $\chi 2$ test.

**Table 2 Comparison of related risk factors in different degrees of AS.**

| Related factors | Normal (n = 2,583) | AS | | | P value |
|---|---|---|---|---|---|
| | | Mild (n = 1,154) | Moderate (n = 589) | Severe (n = 518) | |
| Male n (%) | 1,754 (62.8) | 751 (70.8) | 379 (70.3) | 272 (60.0) | <0.001 |
| Age≥45 years n (%) | 1,452 (52.0) | 844 (79.6) | 482 (89.4) | 443 (97.8) | <0.001 |
| High TC n (%) | 816 (29.2) | 397 (37.5) | 219 (40.6) | 174 (38.4) | <0.001 |
| High TG n (%) | 844 (30.2) | 443 (41.8) | 211 (39.1) | 161 (35.5) | <0.001 |
| High LDL n (%) | 882 (31.6) | 407 (38.4) | 216 (40.1) | 168 (37.1) | <0.001 |
| Low HDL n (%) | 2,699 (96.7) | 1,024 (96.6) | 510 (94.6) | 426 (94.0) | 0.009 |
| Hypertension n (%) | 146 (5.2) | 214 (20.2) | 209 (38.8) | 256 (56.5) | <0.001 |
| Diabetes n (%) | 39 (1.4) | 39 (3.7) | 34 (6.3) | 50 (11.0) | <0.001 |
| High FBG n (%) | 93 (3.3) | 88 (8.3) | 67 (12.4) | 68 (15.0) | <0.001 |
| NAFLD n (%) | 908 (32.5) | 477 (45.0) | 250 (46.4) | 220 (48.6) | <0.001 |
| Current Smoking n (%) | 328 (11.7) | 156 (14.7) | 58 (10.8) | 23 (5.1) | <0.001 |
| Drinking n (%) | 189 (6.8) | 98 (9.2) | 23 (4.3) | 16 (3.5) | <0.001 |
| Exercise n (%) | 428 (15.3) | 81 (7.6) | 17 (3.2) | 14 (3.1) | <0.001 |
| High UA n (%) | 178 (6.4) | 108 (10.2) | 55 (10.2) | 47 (10.4) | <0.001 |

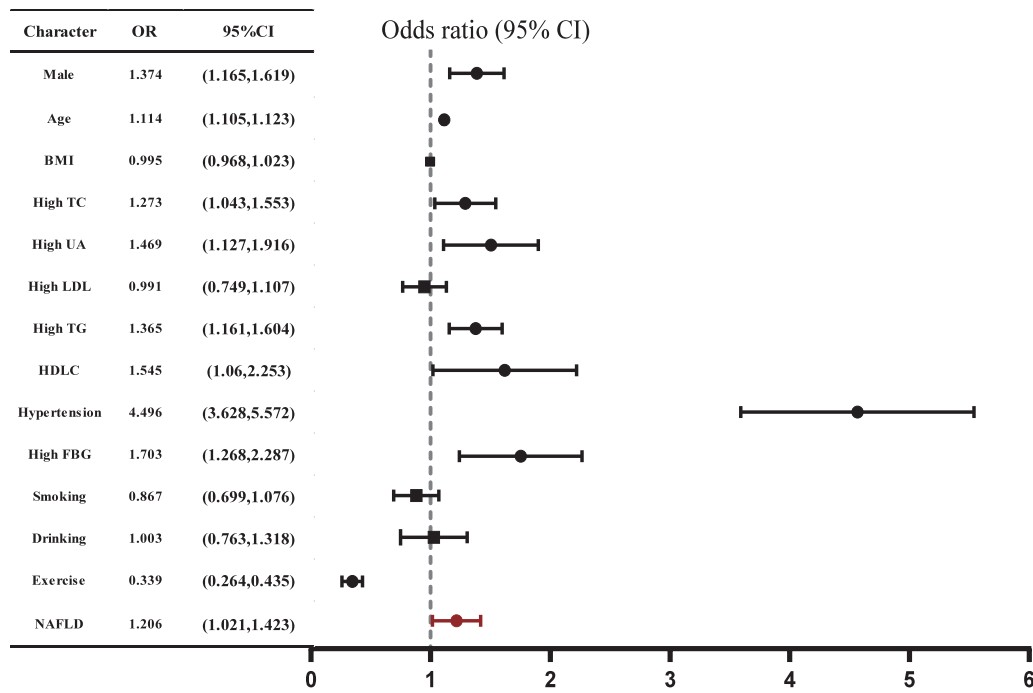

**Figure 2** **Correlation between AS and various risk factors.** Multiple logistic regression analysis was used. $P < 0.05$ in a two-tailed test indicates statistical significance.

with baPWV (Model 2: $\beta = 0.081$, $\Delta R^2 = 0.005$, $P < 0.001$; Model 3: $\beta = 0.052$, $P = 0.001$) (Fig. 4, Table S3). In Model 2 of the women, the positive association between NAFLD and baPWV was greater (Model 2: $\beta = 0.088$, $\Delta R^2 = 0.006$, $P < 0.001$; Model 3: $\beta = 0.042$, $P = 0.02$) (Fig. 4, Table S4).

We also observed an independent positive correlation between NAFLD and baPWV in model 3 of the sensitivity analysis of multiple linear regression models for the whole population ($\beta = 0.050$, $P < 0.001$) (Table S5), males ($\beta = 0.050$, $P = 0.001$) (Table S6), and females ($\beta = 0.043$, $P = 0.014$) (Table S7). In Model 3 of the multiple linear regression analysis of the data set after propensity score matching, NAFLD and baPWV still showed a significant positive correlation ($\beta = 0.040$, $P = 0.006$) (Table S9), and a consistent positive correlation was also maintained in the gender-balanced subsample after random sampling ($\beta = 0.044$, $P < 0.001$) (Table S10).

## Targeted analysis of baPWV in different sub-populations of NAFLD patients

From the age model, post multiple comparison showed that the differences between the groups were significant (Table S8). We observed that baPWV levels in NAFLD patients exhibited a statistically significant elevation compared to non-NAFLD patients across all age categories ($P < 0.001$) (Table 3). Furthermore, the mean baPWV was within the normal arterial elasticity range in NAFLD individuals under 45 years old, while it exceeded the threshold for AS in those over 45 years old. Similarly, in the male subgroup, baPWV began

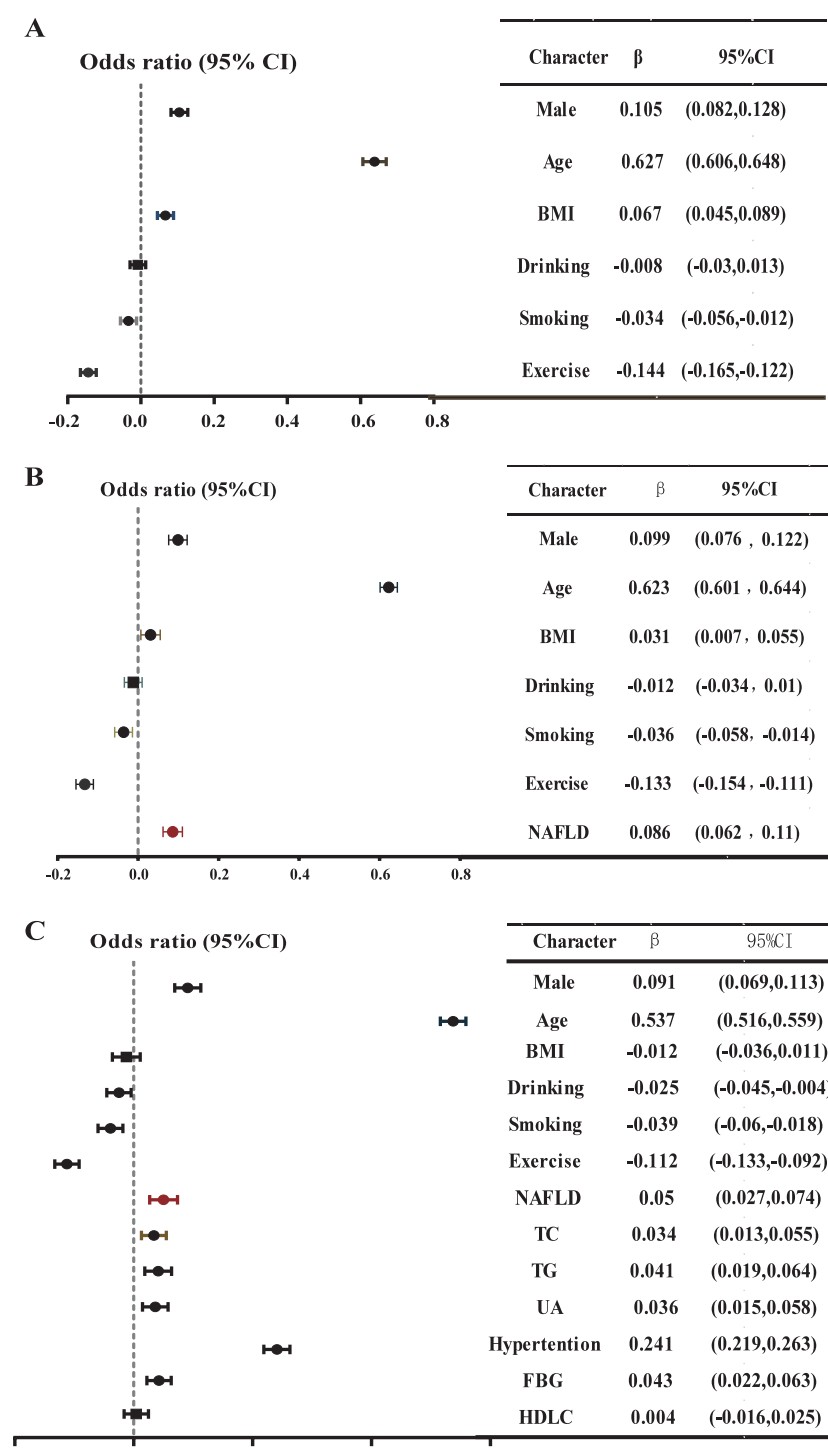

**Figure 3 Multiple linear regression model: Relationship between baPWV and multiple risk factors in the whole population.** (A) Model 1, adjusted for gender, age, BMI, smoking, drinking, and exercise; (B) Model 2, further adjusted NAFLD based on Model 1; (C) further adjusted high TC, high TG, high UA, high FBG, low HDL based on Model 2. A multiple linear regression model was used for analysis. $P < 0.05$ in a two-tailed test indicates statistical significance.

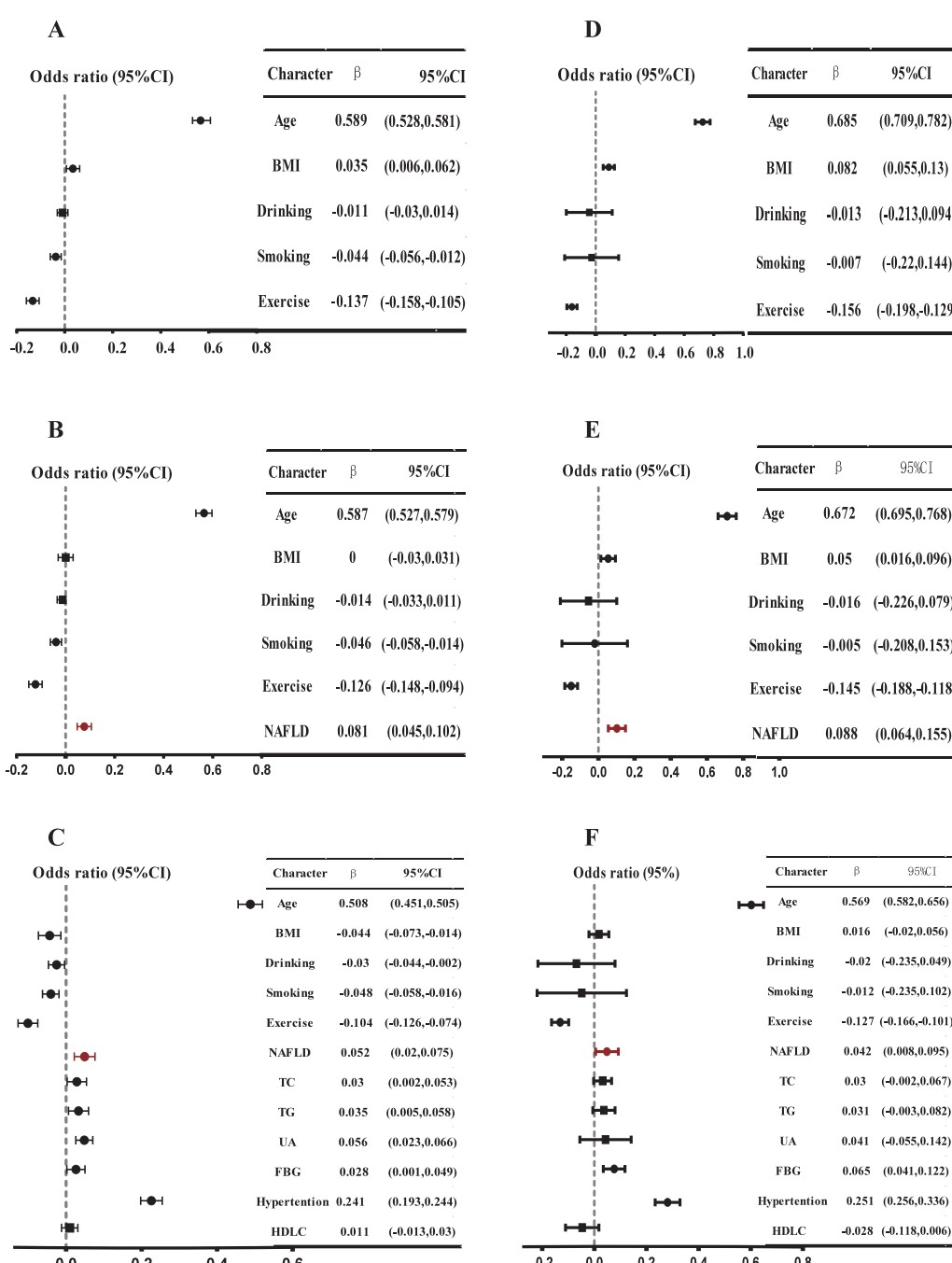

**Figure 4  Multiple linear regression model: Relationship between baPWV and multiple risk factors in different sex.** (A) Model 1, adjusted for age, BMI, smoking, drinking, and exercise in men; (B) Model 2, further adjusted NAFLD based on Model 1 in men; (C) further adjusted high TC, high TG, high UA, high FBG, low HDL based on Model 2. A multiple linear regression model was used for analysis. $P < 0.05$ in a two-tailed test indicates statistical significance in men. (D), (E), and (F) are female data analyses, corresponding to (A), (B), and (C), respectively.

**Table 3  BaPWV in different age groups between NAFLD and non-NAFLD.**

| Age (years) | Total | | NAFLD | | Non-NAFLD | |
|---|---|---|---|---|---|---|
| | $n$ | x ± SD (cm/s) | $n$ | x ± SD (cm/s) | $n$ | x ± SD (cm/s) |
| <35 | 484 | 1,180.66 ± 176.07 | 140 | 1,256.78 ± 172.74 | 344 | 1,149.68 ± 168.03 |
| [35, 45) | 1,139 | 1,273.22 ± 182.61 | 408 | 1,336.22 ± 174.52 | 731 | 1,238.06 ± 177.61 |
| [45, 55) | 1,729 | 1,373.02 ± 201.44 | 728 | 1,419.70 ± 205.51 | 1,001 | 1,339.07 ± 191.50 |
| [55, 65) | 1,011 | 1,523.66 ± 255.48 | 398 | 1,570.03 ± 270.16 | 613 | 1,493.55 ± 240.96 |
| ≥65 | 481 | 1,858.42 ± 362.06 | 181 | 1,938.20 ± 378.94 | 300 | 1,810.30 ± 343.25 |
| Total | 4,844 | 1,409.97 ± 291.24 | 1,855 | 1,471.89 ± 294.90 | 2,989 | 1,371.55 ± 282.25 |
| F | | 763.24 | | 268.07 | | 521.06 |
| P for value[*] | | <0.001 | | <0.001 | | <0.001 |

**Notes.**
According to the above data, baPWV exceeds the normal threshold range when the age of the subjects is over 45 years old.
[*]Trend test of chi-square test; the linear trend test of the polynomial was selected, and $P < 0.05$ was considered to be consistent.

**Table 4  BaPWV in different age groups by gender in NAFLD.**

| Age (years) | Total | | Female | | Male | |
|---|---|---|---|---|---|---|
| | $n$ | x ± SD (cm/s) | $n$ | x ± SD (cm/s) | $n$ | x ± SD (cm/s) |
| <35 | 140 | 1,256.78 ± 172.74 | 19 | 1,106.68 ± 186.96 | 121 | 1,280.35 ± 158.73 |
| (35, 45) | 408 | 1,336.22 ± 174.52 | 57 | 1,287.67 ± 195.31 | 351 | 1,344.10 ± 169.90 |
| (45, 55) | 728 | 1,419.70 ± 205.51 | 134 | 1,375.32 ± 238.56 | 594 | 1,429.71 ± 196.13 |
| (55, 65) | 398 | 1,570.03 ± 270.16 | 137 | 1,590.77 ± 259.40 | 261 | 1,559.15 ± 275.50 |
| ≥65 | 181 | 1,938.20 ± 378.94 | 76 | 1,964.90 ± 355.68 | 105 | 1,918.87 ± 395.48 |
| Total | 1,855 | 1,471.89 ± 294.90 | 423 | 1,527.15 ± 357.08 | 1,432 | 1,455.56 ± 271.84 |
| F | | 268.07 | | 89.57 | | 167.67 |
| P for value[*] | | <0.001 | | <0.001 | | <0.001 |

**Notes.**
According to the above data, baPWV exceeds the normal threshold range when the age of the subjects is over 45 years old.
[*]Trend test of chi-square test; the linear trend test of the polynomial was selected, and $P < 0.05$ was considered to be consistent.

to exceed 1,400 cm/s in the 45–55 age group as well (1,429.71 ± 196.13 cm/s, $P < 0.001$). In the subgroup analysis of women, baPWV exceeded 1,400 cm/s in the 55–65 age group (1,590.77 ± 259.40 cm/s, $P < 0.001$) (Table 4). Postmortem analysis showed that there were significant differences in baPWV among all age groups. And each VIF value is less than 10 in all models. Through trend analysis, the P for trend in all groups was less than 0.001, indicating a linear relationship between baPWV and age.

We proceeded to create age-baPWV scatter plots and fitted lines. In the NAFLD and non-NAFLD groups, the baPWV values reached the diagnostic threshold for AS (1,400 cm/s) at ages 45.62 and 50.8 years, respectively (Fig. 5). In the male and female groups of NAFLD, the baPWV values reached 1,400 cm/s at ages 46.91 and 51.46, respectively. Furthermore, in the sensitivity analysis after filling in the missing values, the baPWV values reached the diagnostic threshold for AS (1,400 cm/s) at ages 45.58 and 50.68 years respectively in the NAFLD and non-NAFLD groups. The baPWV values reached 1,400 cm/s

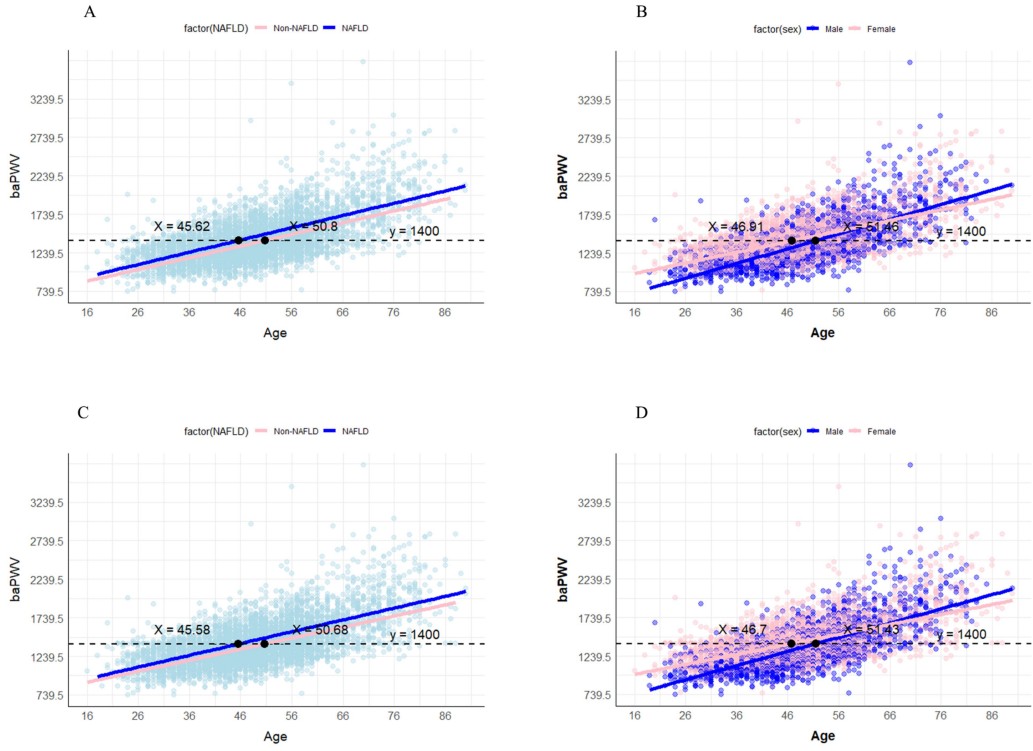

**Figure 5 Age-baPWV scatter plot.** (A) The age-baPWV scatter plot of NAFLD and non-NAFLD groups. (B) the age-baPWV scatter plot of males and females in the NAFLD population; (C) the age-baPWV scatter plot of NAFLD and non-NAFLD groups in sensitivity analysis. (D) the age-baPWV scatter plot of males and females in the NAFLD population in the sensitivity analysis. The number at the intersection of the black dashed line and the two fitted lines indicates the corresponding age when the baPWV reaches 1,400 cm/s.

at ages 46.7 and 51.43 respectively in the male and female groups of NAFLD (Fig. 5). In the analysis of the post-PSM dataset, the age threshold for baPWV to reach the diagnostic threshold of AS (1,400 cm/s) in the NAFLD group was 46.56 years. The age thresholds for men and women were 45.8 and 49.2 years, respectively. In addition, in the gender-balanced dataset, the age threshold corresponding to the NAFLD group was 46.86 years. The age thresholds for male and female groups were 45.4 and 48.7 years, respectively (Fig. S1).

## DISCUSSION

Our study confirmed the correlation between AS and NAFLD by monitoring vascular elasticity using baPWV in a large sample of 4,844 participants. We found that NAFLD was not only significantly positively correlated with baPWV but also was independently associated with higher odds of AS after adjusting for general characteristics and metabolic risk factors associated with ASCVDs. Moreover, baPWV levels increased in the elderly population and met the criteria for AS in NAFLD patients earlier than in non-NAFLD individuals. Through the analysis of age-baPWV scatter plots and fitted lines, combined with sensitivity analysis, it was found that NAFLD patients over the age of 45 have a risk

of baPWV reaching the AS diagnostic threshold (1,400 cm/s). Further refinement suggests that male patients start to have this risk at 46 years old, while female patients at 51 years old. These results provide important guidance for monitoring the occurrence and development of AS in the NAFLD population.

## The relationship between AS, baPWV and NAFLD

Previous studies have progressively revealed the correlation between AS and NAFLD. A cross-sectional study conducted in China demonstrated that NAFLD was independently associated with elevated C-IMT, regardless of conventional cardiovascular and metabolic risk factors, indicating that NAFLD is independently correlated with AS (OR = 1.663, 95% CI [1.391–1.989], $P < 0.0001$) (*Zheng et al., 2018*), which aligns with our findings (*Takeuchi et al., 2012*). There is no pharmacological intervention approved for NAFLD thus far. Studies have proposed preventing AS in NAFLD patients through lifestyle interventions such as the Mediterranean diet and regular exercise (*Wang et al., 2023*). However, reducing the disease burden in NAFLD patients through preventive measures remains largely unrealized (*Eslam, Sanyal & George, 2020*). Therefore, early detection of AS in NAFLD is crucial for timely treatment. Most prospective, retrospective, and genetic studies used vascular ultrasound, CCTA, and CAG to assess arterial health (*Zhang et al., 2023*), measuring atherosclerotic plaques, non-calcified plaques, IMT, and coronary artery calcification scores (CACS) (*Cho et al., 2018*; *Koulaouzidis et al., 2021*; *Lee et al., 2018*; *Wong et al., 2021*). CT, CAG, and invasive imaging including intravascular ultrasound (IVUS) and optical coherence tomography (OCT) involve radiation exposure and the use of iodine contrast media, which carries the risk of renal toxicity and allergic reactions (*Betoko et al., 2021*; *Jinnouchi, Sakakura & Fujita, 2025*; *Miller et al., 2008*). CACS cannot detect non-calcified plaques, which account for 30% of high-risk lesions in the early stages of the disease (*Virmani et al., 2006*). In cases of negative arterial plaque or diffuse arterial thickening, the sensitivity of ultrasound is reduced (*Inaba, Chen & Bergmann, 2012*). However, these procedures require high-quality instrumentation and skilled operators, which may hinder the widespread adoption of these methods in primary hospitals. Additionally, subjective judgments among different operators could potentially introduce errors.

In recent years, a new non-invasive and easy-to-use tool, baPWV, has been gradually introduced into hospitals for monitoring arterial stiffness, facilitating the early identification of AS. It reflects changes in overall arterial elasticity (*Zhu et al., 2015*) and is more cost-effective compared to ultrasound and angiography, especially in medical institutions and regions with limited resources. With the variability of 3.8% to 10.0% within the same operator and 3.6% to 8.4% between different operators, baPWV maintains high repeatability and representativeness (*Munakata, 2014*). While baPWV cannot replace anatomical imaging in assessing localized plaques, its role in early risk stratification is highly valuable. Unlike conventional methods that typically detect atherosclerosis at its later stages, baPWV can signal vascular dysfunction years before structural damage becomes apparent. However, the use of baPWV to investigate the association between NAFLD and AS remains controversial. A prospective study involving 3,433 individuals aged 40 years or older in China revealed a significant and independent association between elevated

baPWV and the development of NAFLD (OR = 1.20; 95% CI [1.07–1.33]; $P = 0.011$), as well as an increased likelihood of fibrosis (*Xin et al., 2020*). A study focusing on NAFLD patients without hypertension or diabetes demonstrated that NAFLD maintained a positive association with baPWV ($\beta = 0.03$, $P = 0.043$) after adjustment for general characteristics, lifestyle factors, blood glucose, lipid profiles, and blood pressure. The effect size was comparable to that of TG and fasting blood glucose on baPWV (*Kim et al., 2012*). Another recent study showed no significant association between NAFLD and baPWV after adjusting for general characteristics and metabolic risk factors (*Bessho et al., 2022*) possibly due to their limited sample size, which included only 890 people, 268 of whom were NAFLD patients.

Our analysis revealed that NAFLD independently elevated baPWV by 30.2 cm/s ($\beta = 0.050$), equivalent to 2.3 years of accelerated vascular aging (calculated using the age-associated coefficient: 13.4 cm/s per year). This hemodynamic impact surpasses that of dyslipidemia (elevated triglycerides: +25.3 cm/s) though remains less pronounced than hypertension (+186.7 cm/s). Critically, the observed 30.2 cm/s elevation translates to an estimated 4.53% increase in cardiovascular mortality risk based on meta-analytic data (15% risk per 100 cm/s) (*Vlachopoulos, Aznaouridis & Stefanadis, 2010*). Consistent with the American College of Cardiology (ACC) clinical practice guidelines recommending appropriate calibration of the 10-year ASCVD risk estimator, NAFLD may potentially be validated as a risk-enhancing factor for adults in the intermediate-risk category (5−7.5% 10-year ASCVD risk) (*Arnett et al., 2019*). Therefore, baPWV measurement may provide clinical utility for early AS screening and risk assessment in NAFLD populations, particularly for patients whose other conventional risk factors (*e.g.*, hypertension, diabetes and dyslipidemia) have been effectively managed. And our identification of age-specific baPWV thresholds further enables personalized vascular surveillance, addressing a critical gap in current NAFLD management guidelines.

## Possible mechanism

Progress has been made in studying the possible mechanisms explaining the causes of AS in NAFLD. The generally accepted mechanisms include lipid metabolism disorders, insulin resistance, systemic inflammatory response, and damage to internal organs and blood vessels caused by oxidative stress (*Zhang et al., 2023*). Our study provides direct evidence supporting metabolic dysregulation as a key pathway linking NAFLD to AS. NAFLD patients exhibited significantly higher levels of FBG ($\beta = 0.043$, $p < 0.001$), TG ($\beta = 0.041$, $p < 0.001$), and UA ($\beta = 0.036$, $p = 0.001$) ,and significant synergistic effects with both hypertension ($\beta = 0.241$, $p < 0.001$) and hypertriglyceridemia in augmenting baPWV compared to non-NAFLD controls, consistent with prior reports (*Deprince, Haas & Staels, 2020*; *Kim et al., 2017*). Despite BMI being non-significant in fully adjusted models ($\beta = -0.012$, $p = 0.311$), metabolic markers remained strongly associated with baPWV, supporting the paradigm shift toward metabolic risk stratification over BMI-based obesity criteria (*Kanwal et al., 2020*; *Wang et al., 2023*; *Younes et al., 2022*).

In addition to the data supporting by our study, external evidence also points to a potential mechanism by which NAFLD affects arteriosclerosis. Mendelian randomization

studies showed NAFLD-driven hepatic insulin resistance promotes lipid overflow into circulation, exacerbating ectopic fat deposition in vascular walls (*Li et al., 2023*). Liver-derived fibroblast growth factor 21 (FGF21) and fetuin-A may directly impair vascular function (*Yafei et al., 2019*). Genetic studies have shown that mutations in specific genes of NAFLD patients may promote abnormal lipid metabolism, heightening the susceptibility to hepatic and vascular fat accumulation and inflammation, thus changing the stiffness of arteries (*Petta et al., 2013*; *Xia et al., 2016*).

There are also emerging hypotheses. While our study did not assess genetic variants, emerging data suggest DNA methylation changes (*e.g.*, CPT1A hypomethylation) may concurrently drive hepatic steatosis and vascular calcification (*Hyun & Jung, 2020*). And NAFLD-associated microbial shifts (*e.g.*, increased Proteobacteria) could elevate circulating trimethylamine N-oxide (TMAO), a known inducer of arterial stiffness (*Shi et al., 2022*; *Zhu et al., 2018*). These mechanisms suggest that NAFLD promotes changes in vascular elasticity and normal functions, which can be manifested by elevated baPWV.

## Targeted assessment of different NAFLD populations

Most studies focus on the incidence of NAFLD and AS in middle-aged and elderly people, but few have elucidated the precise theoretical underpinnings for age selection, and there is a lack of age-stratified analysis of AS occurrence in NAFLD patients. Our study encompassed individuals ranging from 18 to 88 years old, providing a comprehensive exploration of arterial elasticity among NAFLD patients across different age groups. Through the analysis of age-baPWV scatter plots and fitted lines, coupled with sensitivity analysis, it was found that NAFLD patients over the age of 45 face a risk of baPWV reaching the AS diagnostic threshold (1,400 cm/s). Further refinement indicates that the risk starts at 46 years old for male patients and at 51 years old for female patients. Therefore, healthcare providers can facilitate the identification of an increased risk of developing AS in patients with NAFLD by incorporating a patient's gender and age into a holistic assessment. Currently, there is no widely used tool for monitoring arterial conditions in NAFLD patients in clinical practice. Many NAFLD patients only undergo biochemical or imaging tests after symptoms appear, by which time ASCVDs may already be present. This delay in diagnosis hinders early treatment and prevents the timely intervention needed to prevent disease progression. In this study, we explored the feasibility of using baPWV, a non-invasive and convenient tool, for monitoring arterial stiffness in clinical NAFLD patients. This approach may provide valuable insights into the potential of baPWV as a monitoring tool for arterial health in the NAFLD population.

## Application challenges and future directions of baPWV

Our findings advocate a stratified baPWV implementation strategy where clinicians should assess NAFLD patients' general characteristics to guide arterial stiffness monitoring. For primary care, initiate baPWV screening at $\geq 46$ years (males) and $\geq 51$ years (females). NAFLD patients with metabolic comorbidities (hyperuricemia, hypertension, dyslipidemia, or glucose abnormalities) require earlier and biennial monitoring, though optimal intervals need further validation. When baPWV exceeds the diagnostic threshold for

AS (1,400 cm/s), it should prompt immediate reassessment of cardiovascular disease risk. Based on the patient's overall health condition, determine whether additional imaging or pharmacological interventions are needed to guide referral decisions. In primary care, portable baPWV devices can serve as a first-line screening tool for arterial stiffness. To enhance clinical utility and conduct early intervention and optimize cardiovascular risk management, exploring the integration of AI-enhanced portable baPWV devices could improve accessibility. Specialized hospitals, equipped with advanced resources and facilities, should prioritize developing an ASCVD risk prediction model that combines baPWV thresholds with clinical parameters such as liver fibrosis (FIB-4 index) to improve diagnostic accuracy. Additionally, baPWV measurements can serve as a stratification tool to guide the necessity of advanced imaging.

Several challenges must be addressed for real-world implementation. The generalizability of age-baPWV risk thresholds requires validation in diverse populations, as genetic, ethnic, and regional differences in vascular aging may alter these cutoffs. Our study showed a significant male predominance in the NAFLD cohort. Our analysis using sex-subgroup, propensity score matching, and sex-balanced sub-samples confirmed a significant positive correlation between NAFLD and baPWV. Nevertheless, residual confounding from unmeasured sex-specific factors (*e.g.*, hormonal profiles, body fat distribution) cannot be fully excluded (*Lazo et al., 2015*). Consequently, our conclusions may be more applicable to male NAFLD patients, and extrapolation to females requires caution. Future sex-stratified longitudinal studies should clarify gender-specific NAFLD-CVD mechanisms, while public health initiatives enhance NAFLD screening in underrepresented female populations to refine risk stratification strategies. Cost-effectiveness analyses are needed to justify large-scale deployment of baPWV devices in community health programs, especially in low-income regions where competing health priorities exist. BaPWV $\geq$ 1,400 cm/s, while indicative of arterial stiffness, may not exclusively represent AS pathology; it could also capture functional vascular changes from transient conditions (*e.g.*, acute inflammation or hypertension episodes).

The main advantage of this study is that it has a large sample size and a population-based cohort with favorable characteristics. Furthermore, large-sample analyses in male and female NAFLD patients were conducted to evaluate the significance of using baPWV to monitor AS in NAFLD patients at different ages. While baPWV may be less accurate than percutaneous coronary angiography and ultrasonic in diagnosing AS, it is a simpler, more convenient and cost-effective tool. The clinical and biochemical indicators we selected to participate in the risk assessment are also easily accessible and applicable in basic medical institutions, providing methodological guidance for areas with limited medical resources.

However, the cross-sectional nature of the study limits the ability to establish a causal relationship between NAFLD and AS, necessitating further prospective research for confirmation. Additional investigations are required to clarify the exact biological pathways mediating this relationship and to establish whether NAFLD directly contributes to atherosclerotic pathogenesis or simply reflects concurrent metabolic disturbances. Ultrasound's limited sensitivity in detecting mild steatosis (<20% hepatic fat) and in obese populations may lead to underdiagnosis, particularly in high-BMI NAFLD patients

(*Paige et al., 2017*). This limitation introduces selection bias, excluding obese individuals (BMI > 30 kg/m$^2$) and those with mild steatosis, affecting the generalizability of our findings. To address this, we plan to collect clinical data from biopsy-confirmed NAFLD cases, focusing on obese patients and those with mild steatosis for future targeted analyses. Additionally, the study's male–female subject disparity, qualitative assessment of fatty liver, and single-location focus in Wuhan may introduce bias and limit generalizability. Although we adjusted for major metabolic confounders, residual confounding by unmeasured factors (*e.g.*, dietary patterns) cannot be excluded. Since the study population we included was older on average and more male, these could have contributed to a higher proportion of abnormal HDL. Mediating analysis has not been performed to further explore the influence of various factors on baPWV. In the future, more valuable confounding indicators need to be screened and included, and efforts should be made to carry out mediating analysis actively. In addition, this study revealed the value of baPWV combined with age and gender in monitoring the occurrence of AS in NAFLD patients, but further positive verification of the model was lacking. Therefore, we will need to collect more data in the future to verify the model of this study. Future research should aim for multi-regional and multi-institutional collaboration and conduct more detailed sensitivity studies to develop comprehensive evaluation programs.

## CONCLUSION

BaPWV is a simple and economical non-invasive means for the early monitoring of arterial elasticity in individuals with NAFLD. People with NAFLD have an increased risk of AS after adjusting basic clinical characteristics and conventional metabolic risk factors associated with ASCVDs. NAFLD also exhibited an independent positive correlation with elevated baPWV, which can be attributed to the alterations in arterial elasticity and dysfunction induced by NAFLD through diverse mechanistic pathways. Patients with NAFLD aged over 45 years should undergo regular monitoring for AS using baPWV. Furthermore, it is recommended that male patients initiate monitoring at 46 years of age, whereas female patients should commence monitoring at 51 years of age.

**Abbreviations**

| | |
|---|---|
| **ASCVDs** | atherosclerotic cardiovascular diseases |
| **AS** | atherosclerosis |
| **AT** | Acti Test |
| **baPWV** | brachial-ankle pulse wave velocity |
| **ACC** | American College of Cardiology |
| **BMI** | body mass index |
| **CV** | cardiovascular |
| **CVD** | cardiovascular disease |
| **CCTA** | coronary computed tomography angiography |
| **CAG** | coronary angiography |
| **CACS** | coronary artery calcification scores |
| **FT** | Fibro Test |

| FBG | fasting blood glucose |
| FGF21 | fibroblast growth factor 21 |
| HDL | high-density lipoprotein cholesterol |
| IMT | intima-media thickness |
| IQR | interquartile range |
| IVUS | intravascular ultrasound |
| IRS-1 | insulin receptor substrate 1 |
| LDL | low-density lipoprotein |
| NAFLD | non-alcoholic fatty liver disease |
| NAFL | non-alcoholic fatty liver |
| OCT | optical coherence tomography |
| ST | Steato Test |
| SMD | standardized mean differences |
| TG | triglyceride |
| TC | total cholesterol |
| TMAO | trimethylamine N-oxide |
| UA | uric acid |
| VIF | variance inflation factors |

## ACKNOWLEDGEMENTS

We are especially grateful to all the subjects who participated in this study at the Physical Examination Center of Zhongnan Hospital of Wuhan University.

### Funding

This work was supported by the Hospital Discipline Capacity Building Project of Hubei Provincial Department of Finance (YYXKNLJS2024013), Hubei Provincial Natural Science Foundation of China (2024AFB797) and the project of Hubei Provincial Health Commission—the establishment of big data of patients with slow physical examination and the application of Internet + health management model (WJ2023F011). The funders had no role in study design, data collection and analysis, decision to publish, or preparation of the manuscript.

### Grant Disclosures

The following grant information was disclosed by the authors:
The Hospital Discipline Capacity Building Project of Hubei Provincial Department of Finance: YYXKNLJS2024013.
Hubei Provincial Natural Science Foundation of China: 2024AFB797.
The project of Hubei Provincial Health Commission—the establishment of big data of patients with slow physical examination and the application of Internet + health management model: WJ2023F011.

## Competing Interests

The authors declare there are no competing interests.

## Author Contributions

- Yujie Wang performed the experiments, analyzed the data, prepared figures and/or tables, and approved the final draft.
- Zhicheng Fang performed the experiments, analyzed the data, prepared figures and/or tables, and approved the final draft.
- Qiuyue Fu performed the experiments, prepared figures and/or tables, and approved the final draft.
- Dongai Yao conceived and designed the experiments, performed the experiments, authored or reviewed drafts of the article, and approved the final draft.
- Xiaoqing Jin conceived and designed the experiments, performed the experiments, prepared figures and/or tables, authored or reviewed drafts of the article, and approved the final draft.

## Human Ethics

The following information was supplied relating to ethical approvals (i.e., approving body and any reference numbers):

The study was approved by the ethics committee of the Zhongnan Hospital of Wuhan University (NO.2021114) and it was granted an exemption from informed consent.

## Clinical Trial Ethics

The following information was supplied relating to ethical approvals (i.e., approving body and any reference numbers):

The study was approved by the ethics committee of the Zhongnan Hospital of Wuhan University (NO.2021114) and it was granted an exemption from informed consent. The study was carried out in accordance with the applicable guidelines and regulations.

## Data Availability

The original data is available in the Supplementary File.

## Clinical Trial Registration

The following information was supplied regarding Clinical Trial registration:

NO. 2021114.

## Supplemental Information

Supplemental information for this article can be found online at http://dx.doi.org/10.7717/peerj.19405#supplemental-information.

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
