# Peer review of "Association between non-alcoholic fatty liver disease and arterial stiffness measured by brachial-ankle pulse wave velocity: a cross-sectional population study"

_PeerJ, doi:10.7717/peerj.19405_

## Round 0.1 · original submission · Major Revisions

Please revise your manuscript according to the reviewers' comments.

Yours,
Yoshi
Prof. Yoshinori Marunaka, M.D., Ph.D.

Reviewer 1 ·

Basic reporting

For each of the grouping models, results should be shown in the form of figures in addition to the tables, to increase the ease of data visualization.

Experimental design

A major drawback of this study is that NAFLD grouping was done by ultrasound operators without further validation by clinicians. There are several reports that operators are not as efficient at recognizing disease characteristics as clinicians. For both the ultrasounds used to confirm NAFLD as well as BaPWV measurements, validation by a clinician is necessary to increase robustness of data and inferences. If this has not been done, it should be in the discussion as a limitation of the study.

Validity of the findings

no

Additional comments

Line 168 – percentage of men is 77.2% of all NAFLD patients.
Line 169 – percentage of women is 22.8% of all NAFLD patients.

Reviewer 2 ·

Basic reporting

Manuscript ID: (2024:05:101147:0:4:NEW)
Title: "Correlation between non-alcoholic fatty liver disease and arteriosclerosis: A population-based study"

Dear Authors,
I have thoroughly reviewed your manuscript. While your study addresses an important topic, several major concerns need to be addressed before publication can be considered.
MAJOR STRUCTURAL CONCERNS:
1. Title and Abstract:
Major Revisions Required:
- Current title doesn't reflect the specific methodology (baPWV)
- Suggested revision: "Association between non-alcoholic fatty liver disease and arterial stiffness measured by brachial-ankle pulse wave velocity: A cross-sectional population study"

Abstract needs substantial revision:
- Missing confidence intervals for main effects
- No mention of study limitations
- Lacks specific diagnostic criteria for NAFLD and AS
- Statistical methods insufficiently described
- Power calculations not mentioned
- Missing handling of potential confounders

2. Introduction (Lines 60-91):
Critical Weaknesses:
a) Literature Review:
- Inadequate presentation of existing knowledge
- Missing systematic reviews and meta-analyses
- Incomplete citations (e.g., "EWaiSun & Mary 2021")
b) Research Gap:
- Poorly defined controversy regarding NAFLD-AS relationship
- Inadequate justification for choosing baPWV over traditional methods
3. Methodology (Lines 93-164):
Major Deficiencies:
a) Study Design:
- Missing sample size calculation
- No power analysis
- Unclear handling of missing data
- No mention of quality control measures
b) Measurements:
- Insufficient detail on ultrasound protocol
- Missing standardization procedures
- Laboratory methods inadequately described
- BaPWV measurement conditions not specified
c) Statistical Analysis:
- No mention of normality testing
- Missing outlier handling
- Multiple comparison adjustments not addressed
- Effect size calculations absent

4. Results Validity and Novelty (Lines 166-207):
Critical Issues:
a) Data Plausibility:
- Unusually high HDL abnormality (96.2%) requires explanation
- Missing sensitivity analyses
- No cross-validation of findings
- R² values not reported for regression models
b) Novelty Concerns:
- Primary findings largely confirmatory of existing literature
- No innovative biomarkers or parameters
- Limited advancement beyond existing knowledge
- Similar findings to Zhuojun et al. 2020

5. Statistical and Analytical Concerns:
a) Model Building:
- Insufficient justification for variable selection
- Missing interaction analyses
- No assessment of model assumptions
- Limited exploration of confounding effects
b) Missing Analyses:
- No mediation analysis
- Lack of sensitivity testing
- Missing subgroup analyses
- No validation cohort
6. Discussion Section Issues (Lines 208-323):
Major Problems:
a) Over-interpretation:
- Causal language inappropriate for cross-sectional design
- Speculation about mechanisms without supporting data
- Inadequate acknowledgment of limitations
- Overstatement of clinical utility
b) Mechanistic Claims:
- SREBP-1c discussed without measurement
- Blood viscosity mentioned without data
- Mitochondrial activity claims unsupported
- Pathophysiological pathways speculative
c) Age-Related Findings:
- Arbitrary selection of 45-year threshold
- No formal interaction testing
- Missing sensitivity analyses for age cut-points
- Limited justification for chosen threshold
7. Conclusion Misalignment (Lines 324-333):
Significant Issues:
- Causal claims inappropriate for study design
- Risk population recommendations premature
- Implementation guidance lacking evidence base


Required Text Revisions:
a) Abstract:
Current: "NAFLD increased AS odds by 20.6% (P=0.027)"
Revise to: "NAFLD was associated with increased odds of AS (OR: 1.206, 95% CI: 1.021-1.423, P=0.027) after adjusting for confounders"
b) Methods Section:
Add mandatory subsections:
- Sample Size Calculation
- Quality Control Measures
- Statistical Power Analysis
- Missing Data Handling
- Reproducibility Assessment

c) Results Section:
Add essential elements:
- Model diagnostics
- Sensitivity analyses results
- Subgroup analyses
- Effect modifications
- Complete confounding assessment

13. Language and Terminology Corrections:

a) Replace Causal Language:
- "NAFLD was an independent risk factor" → "NAFLD was independently associated with"
- "increased AS risk" → "was associated with higher odds of AS"
- "predicted AS" → "was associated with AS"

b) Statistical Terminology:
- Add precise definitions of:
* Risk factor categorization
* Outcome measurements
* Statistical approaches
* Effect size interpretations

- Add forest plots
- Include ROC curves
- Present correlation matrices
- Add residual plots
- Include flow diagrams

Discussion:
- Remove causal claims
- Add limitations
- Include implementation challenges
- Discuss alternative explanations
- Add future directions

The manuscript requires substantial revision before it can be considered for publication. All these changes are necessary to meet the scientific standards of the journal.

Experimental design

all comments are listed in the basic report

Validity of the findings

all comments are listed in the basic report

Reviewer 3 ·

Basic reporting

The article Correlation between non-alcoholic fatty liver disease
and arteriosclerosis : A population-based study represents an interesting research that included a large number of participants, making it more valuable.
The article is written in understandable manner and is easy to follow. The background data in introduction fully support the research.
NAFLD surely represents a high risk for CVDs, and is necessary to select patients that are at higher risk even in early stage of NAFLD.

Experimental design

Study is well designed, reproducible, and the criteria for patients selection are valid. Numerous risk factors were included in consideration to find as much as possible correlations with NAFLD and AS.
Question for authors:
Are there any additional data like ultrasound that can confirm the presence or absence and the stage of atherosclerosis regardless baPWV method?
Provide some more references that confirm the validity of baPWV for atherosclerosis confirmation.

Validity of the findings

Findings of the study can be very useful and it is important to check patients with NAFLD over 45 age for arteriosclerosis and CVDs risk.
These findings are significant.
Are there any limitations for this risk assessment? If so, please provide in the text.

Additional comments

In the title, atherosclerosis should be replaced with brachial-ankle pulse wave velocity, since this is an indirect method to confirm the presence of AS, but not gold standard for AS diagnosis.

Also, some tables could be replaces with graph chart, in order to point out the major findings.

---

## Round 0.2 · Minor Revisions

Please submit your manuscript revised according to the reviewer's comments.
Yours,
Prof. Yoshinori Marunaka, M.D., Ph.D.

Reviewer 2 ·

Basic reporting

Dear Authors,
Thank you for your extensive revisions to the manuscript "Association between non-alcoholic fatty liver disease and arterial stiffness measured by brachial-ankle pulse wave velocity: A cross-sectional population study." I have carefully reviewed both your original manuscript and your detailed response to the reviewers' comments. You have made commendable efforts to strengthen the methodological rigor, statistical analysis, and presentation of your findings.
The manuscript has been substantially improved in several key areas:

The revised title and abstract now appropriately reflect the methodology and include important statistical details such as confidence intervals and effect sizes.

The enhanced statistical analysis sections with normality testing, outlier handling, VIF assessment, sensitivity analyses, and gender-specific subgroup analyses strengthen the reliability of your findings.

The inclusion of a detailed sample size calculation demonstrates methodological rigor and transparency.
The replacement of arbitrary age thresholds with data-driven determinations through scatter plots and fitted lines provides more evidence-based clinical recommendations.

The revised language throughout the manuscript appropriately avoids implying causality from cross-sectional data.

However, to further enhance the scientific merit and clinical utility of your work, I recommend addressing the following remaining issues:

1. NAFLD Diagnostic Validation

Acknowledge more explicitly the limitations of ultrasound-based diagnosis compared to liver biopsy.
Consider providing any available data on the sensitivity and specificity of your diagnostic approach
Discuss how potential misclassification might affect your findings

2. Gender Imbalance Analysis
The significant gender imbalance in your NAFLD cohort (77.2% men vs. 22.8% women) requires more substantive statistical consideration:

Consider implementing propensity score matching or weighting to better account for this imbalance.
Provide a more extensive discussion of how this imbalance might affect the generalizability of your findings to the broader NAFLD population.
Consider presenting a sensitivity analysis using a gender-balanced subsample if possible.

3. Effect Size Clinical Relevance
Your multiple linear regression analysis found a statistically significant but relatively modest effect size (β=0.050) for the association between NAFLD and baPWV:

Consider contextualizing this effect size relative to other established risk factors for arterial stiffness.
Discuss the potential implications of this effect size for risk stratification and clinical decision-making.

4. Mechanistic Discussion Enhancement:

More explicit connections between the proposed mechanisms and your observed associations.
Clearer acknowledgment of which mechanisms are directly supported by your data versus those that remain speculative.
A more structured presentation distinguishing between established pathways and emerging hypotheses.

5. Clinical Implementation Guidance
Your section on "Application challenges and future directions of baPWV" would be strengthened by:

More concrete recommendations for how clinicians should interpret and act upon baPWV measurements in NAFLD patients
A clearer algorithm or decision pathway for integrating baPWV into cardiovascular risk assessment for NAFLD patients
Discussion of how your findings might be implemented in different clinical settings (primary care vs. specialty clinics)

Experimental design

no comment

Validity of the findings

no comment

Additional comments

no comment

Reviewer 3 ·

Basic reporting

Based on suggestions and comments, authors have significantly improved the article and it can be published.

Experimental design

Based on suggestions and comments, authors have significantly improved the article and it can be published.

Validity of the findings

Based on suggestions and comments, authors have significantly improved the article and it can be published.

Additional comments

Based on suggestions and comments, authors have significantly improved the article and it can be published.

---

## Round 0.3 · accepted · Accept

Congratulations!
Yours,
Yoshi
Prof. Yoshinori Marunaka, MD., Ph.D.